# Male-Dominant Hepatitis A Outbreak Observed among Non-HIV-Infected Persons in the Northern Part of Tokyo, Japan

**DOI:** 10.3390/v13020207

**Published:** 2021-01-29

**Authors:** Masayuki Honda, Hiroyuki Asakura, Tatsuo Kanda, Yoshiko Somura, Tomotaka Ishii, Yoichiro Yamana, Tomohiro Kaneko, Taku Mizutani, Hiroshi Takahashi, Mariko Kumagawa, Reina Sasaki, Ryota Masuzaki, Shini Kanezawa, Kazushige Nirei, Hiroaki Yamagami, Naoki Matsumoto, Mami Nagashima, Takashi Chiba, Mitsuhiko Moriyama

**Affiliations:** 1Division of Gastroenterology and Hepatology, Department of Medicine, Nihon University School of Medicine, Itabashi, Tokyo 173-8610, Japan; honda.masayuki@nihon-u.ac.jp (M.H.); ishii.tomotaka@nihon-u.ac.jp (T.I.); yamana.yoichiro@nihon-u.ac.jp (Y.Y.); kaneko.tomohiro@nihon-u.ac.jp (T.K.); mizutani.taku@nihon-u.ac.jp (T.M.); hiroshi.t.215@gmail.com (H.T.); kumagawa.mariko@nihon-u.ac.jp (M.K.); sasaki.reina@nihon-u.ac.jp (R.S.); masuzaki.ryota@nihon-u.ac.jp (R.M.); kanezawa.shini@nihon-u.ac.jp (S.K.); nirei.kazushige@gmail.com (K.N.); yamagami.hiroaki@nihon-u.ac.jp (H.Y.); matsumoto.naoki@nihon-u.ac.jp (N.M.); moriyama.mitsuhiko@nihon-u.ac.jp (M.M.); 2Department of Microbiology, Tokyo Metropolitan Institute of Public Health, Shinjuku, Tokyo 169-0073, Japan; Hiroyuki_Asakura@member.metro.tokyo.jp (H.A.); Yoshiko_Soumura@member.metro.tokyo.jp (Y.S.); Mami_Nagashima@member.metro.tokyo.jp (M.N.); Takashi_Chiba@member.metro.tokyo.jp (T.C.)

**Keywords:** male-dominant, men who have sex with men, HAV, non-HIV, sexually transmitted diseases

## Abstract

Recently, we experienced an outbreak of acute hepatitis A virus (HAV) infection between 2018 and 2020. Herein, we describe this male-dominant HAV infection outbreak observed among non-human immunodeficiency virus (HIV)-infected persons in the northern part of Tokyo, Japan. Clinical information was collected from patient interviews and from medical record descriptions. In the present study, 21 patients were retrospectively analyzed. A total of 90.4 and 33.3% of patients were males, and men who have sex with men (MSM), respectively. The total bilirubin levels and platelet counts tended to be lower in the MSM group than in the non-MSM group. C-reactive protein (CRP) levels tended to be higher in acute liver failure (ALF) patients than in non-ALF patients. Prolonged cholestasis was observed in one patient (4.8%). We also found that 18 HAV isolates belonged to HAV subgenotype IA/subgroup 13 (S13), which clustered with the HAV isolate (KX151459) that was derived from an outbreak of HAV infection among MSM in Taiwan in 2015. Our results suggest that the application of antivirals against HAV, as well as HAV vaccines, would be useful for the treatment and prevention of severe HAV infection.

## 1. Introduction

Hepatitis A virus (HAV), a positive-sense, single-stranded RNA virus, is a major cause of acute hepatitis and acute liver failure worldwide, which occasionally leads to liver transplantation or death [1,2,3,4]. Recently, HAV was found to have two virus forms—a membrane-cloaked quasi-enveloped virus (eHAV), and a naked, nonenveloped virion—that exist in the blood circulation and feces, respectively, of patients with acute hepatitis A [5,6].

HAV usually infects humans through the fecal–oral route via contaminated food and water. Because of improved sanitary and socioeconomic conditions, as well as the introduction of the HAV vaccination, the number of patients with acute HAV infection has declined, especially in developed countries [1]. In Japan, as there are no universal vaccination programs against HAV, susceptibility to HAV infection has increased in the general population [7,8].

It is well known that men who have sex with men (MSM) are a high-risk group for HAV infection [9,10,11]. After 1998, sexual activity was one of the most important transmission routes of HAV infection in the metropolitan Tokyo area of Japan [12]. Koga et al. analyzed the clinical features of HAV infection in people living with human immunodeficiency virus (HIV) between pandemics in 1999–2000 and 2017–2018 in the metropolitan Tokyo area of Japan [13]. Of interest, all 16 HAV–HIV-coinfected patients (five and 11 in 1999–2000 and 2017–2018, respectively) were MSM [13].

In the present study, we analyzed 21 patients with acute hepatitis A who were admitted to our department of the Nihon University School of Medicine Itabashi Hospital, in the northern part of Tokyo, Japan, between January 2018 and April 2020. Of note, all these patients were HIV-negative; however, we recognized hepatitis A as a male-dominant disease, and identification as an MSM as one of the high-risk factors for HAV infection. These characteristics seem to be recent trends in HAV infections in the Tokyo area of Japan.

## 2. Patients and Methods

### 2.1. Patients

A total of 21 patients with acute hepatitis A who were admitted to Nihon University School of Medicine Itabashi Hospital, Itabashi-ku, Tokyo, Japan between 1 January 2018 and 30 April 2020 were analyzed in the present study. Acute hepatitis A was diagnosed by positive testing for immunoglobulin M (IgM) antibodies to HAV. All participants were negative for the IgM anti-hepatitis B virus (HBV) core antibody, the anti-hepatitis C virus (HCV) antibody, the IgA anti-hepatitis E virus (HEV) antibody, or for anti-HIV antibody. This retrospective study was approved by the ethics committee of Nihon University School of Medicine Itabashi Hospital (protocol no. RK-180911-12). Participation in the study was posted at the website of our institution. This study conformed to the ethical guidelines of the Declaration of Helsinki.

### 2.2. Clinical and Laboratory Assessments

Clinical parameters were measured by standard laboratory techniques. When the patients were diagnosed with acute hepatitis A, the attending doctor contacted the staff of Itabashi-ku, Tokyo, and analyzed serum HAV RNA. Because hepatitis A is an infectious disease, by law, a physician must immediately report a diagnosis in Japan to the authorities. Clinical information was collected by patient interviews, and all data were collected from medical record descriptions.

### 2.3. Definition and Classification of Acute Liver Failure

According to Japanese criteria, patients showing prothrombin time values of 40% or less of the standard value, or international normalized ratios (INRs) of 1.5 or more, due to severe liver damage within 8 weeks of the onset of disease symptoms are diagnosed as having “acute liver failure”, where liver function prior to the current onset of liver damage is estimated to have been normal based on blood laboratory data and imaging examinations. “Acute liver failure” (ALF) is categorized into two forms, “ALF without hepatic coma” or “ALF with hepatic coma” [14].

### 2.4. Detection of HAV RNA, Sequence Analysis and Phylogenetic Tree Analysis of HAV Isolates

Sera were collected by Itabashi City Health Center, and analysis of HAV RNA was performed by the Department of Microbiology, Tokyo Metropolitan Institute of Public Health, Shinjuku, Tokyo, Japan when the patients were diagnosed with acute hepatitis A. Total RNA were extracted from 140 μL serum samples using a QIAamp Viral RNA Mini Kit (Qiagen, Tokyo, Japan) and subjected it to RT-PCR for the amplification of HAV VP1-2A region [15,16,17]. Complementary DNA (cDNA) was synthesized with primer HAV-JCT-1R-A (YTTRTCATCYTTCATTTCTGTCCA) at 56 °C for 45 s, 50 °C for 40 min and 95 °C for 15 min. Then cDNA was amplified with primers HAV-JCT-1R-A and HAV-JCT-2F (GRAGAACAGGRAAYATTCARATTAG) with 45 cycles of 94 °C for 1 min, 53 °C for 30 s and 72 °C for 2 min, followed by a single cycle of 10 min at 72 °C and 1 min at 20 °C, using QUIAGEN OneStep RT-PCR Kit (Qiagen) in a GeneAtlas Thermal Cycler (Astec, Tokyo, Japan). Then, first PCR product was amplified further with primer HAV-JCT-2R (CAGTHARMACHCCAGCATCCAT) and HAV-JCT-2F with denaturation of 94 °C for 3 min and 35 cycles of 94 °C for 1 min, 58 °C for 2 min and 72 °C for 2 min, followed by a single cycle of 6 min at 72 °C and 45 s at 20 °C [17].

Direct sequencing of the amplified product was performed with primers previously reported [15,16,17]. The HAV VP1-2A region was determined and deposited in the DDBJ/EMBL/GenBank databases under accession numbers LC597557 to LC597576. A phylogenetic tree was constructed by the neighbor-joining (N-J) method [17]. To confirm the reliability of the phylogenetic tree, bootstrap resampling tests were performed 1000 times. These analyses were performed using the Molecular Evolutionary Genetic Analysis (MEGA7) software system (http://www.megasoftware.net/).

### 2.5. Statistical Analysis

Data are expressed as the mean ± standard deviation (SD). Statistical analyses were performed using Student’s *t*-test or a chi-squared test. Variables with *p* < 0.05 in the univariate analyses were evaluated using multiple logistic regression analysis. The results were considered statistically significant at *p* < 0.05. Statistical analysis was conducted using DA stats software version PAF01644 (NIFTY Corp., Tokyo, Japan) and the Excel Statistics program for Windows 2010 (SSRI, Tokyo, Japan).

## 3. Results

### 3.1. Patient Characteristics and Comparison of ALF and Non-ALF Patients

The baseline characteristics of the patients with acute hepatitis A are shown in Table 1. All 21 patients were positive for the IgM anti-HAV antibody, but negative for the HBsAg or anti-HIV antibodies. We performed univariate analyses and compared the clinical characteristics of ALF patients with those of non-ALF patients (Table 1 and Table 2). In the present study, there were no cases of coma amongst the ALF patients. The aspartate aminotransferase (AST), alanine aminotransferase (ALT) and lactate dehydrogenase (LDH) levels were higher in ALF patients than in non-ALF patients. Of interest, C-reactive protein (CRP) levels tended to be higher in ALF patients than in non-ALF patients. The duration from onset to admission tended to be shorter in ALF patients than in non-ALF patients. Prolonged cholestasis was found in one (4.8%) of the 21 patients in the present study (case no. 20 in Table 2). However, the multivariate analysis did not find factors contributing to ALF, as the study number was too small.

### 3.2. Comparison of MSM with Non-MSM Groups’ Characteristics

Patient lists are shown in Table 2. Among the 21 patients, two and three patients had recent histories of eating raw oysters or raw fish, respectively. A total of four and three patients had histories of syphilis and hepatitis B, respectively. Of interest, 19 (90.4%) and seven (33.3%) of the 21 patients were male and MSM, respectively. A total of seven (36.8%) of the 19 male patients were MSM (Table 1 and Table 2). Of the two female patients, one spent time in the MSM community.

We compared the baseline characteristics of the MSM group with those of the non-MSM group with acute hepatitis A via univariate analyses. The total bilirubin levels and platelet counts tended to be lower in the MSM group than in the non-MSM group (Table 3).

### 3.3. Factors of Administration for 20 Days or More in Japanese Patients with Acute Hepatitis A

We also compared the baseline characteristics of patients with admission for less than 20 days and those with admission for 20 days or more via univariate analyses (Table 4). Patients with admission for 20 days or more tended to have higher AST levels or γ-glutamyl transpeptidase (γ-GTP) levels than those with admission for less than 20 days. However, the multivariate analysis did not find the factors contributing to ALF, as the study sample size seemed to be too small.

### 3.4. Molecular Analysis of HAV Isolates from Some Patients with Acute Hepatitis A in the Present Study

We recorded 15 and six patients with acute hepatitis A in 2018 and in the period January 2019 to April 2020, respectively. The HAV VP1-2A region from the 20 isolates derived from 21 of these patients was analyzed in a phylogenetic tree based on the neighbor-joining method (Figure 1). All 20 isolates belonged to the HAV subgenotype IA. Among them, only two isolates were similar to subgroup 9 (S9) [17], and the other 18 isolates clustered (subgroup 13 (S13)) with the HAV isolate (KX151459), which was identified following an outbreak of HAV infection among MSM in Taiwan in 2015 [15] (Figure 1).

## 4. Discussion

As sanitation has improved and susceptibility to HAV infection has increased in the general population [7], it seems that outbreaks of hepatitis A can be expected to occur in the near future in Japan [18]. The number of cases of acute hepatitis A was 100–300 annually between 2001 and 2017 in Japan. In 2018, 925 patients with acute hepatitis A were reported [19]. In our institute, we also identified 15 patients with acute hepatitis A and no HIV infection in 2018. We analyzed 21 patients that were identified between January 2018 and April 2020. We also confirmed that the HAV isolates clustered with the HAV isolate (KX151459) derived from the outbreak of HAV infection among MSM in Taiwan in 2015.

We found that AST, ALT, and LDH levels were higher in the five ALF patients than in the 16 non-ALF patients. We also observed that CRP levels tended to be higher in the ALF patients than in the non-ALF patients. Patients with acute hepatitis A often display fever [20]. The present study also demonstrated that 13 (61.9%) of the 21 patients had a fever (temperature equal to or higher than 38 °C) at admission. In the present study, we found prolonged cholestasis in 4.8% of patients. Jung et al. reported that prolonged cholestasis was found in 4.7% of patients in Korea [21], and could be predicted from preexisting chronic HBV infection, prolonged prothrombin time, and higher total bilirubin levels. Our patient had higher total bilirubin, but no preexisting chronic HBV infection or prolonged prothrombin time (case no. 20 in Table 2).

Our previous study of a hepatitis A outbreak associated with a revolving sushi bar in Chiba, Japan, in 2011 demonstrated that 11 (40.7%) of the 27 patients were male, indicating that the patients were not male-dominant [22]. Further, in an HAV infection study in metropolitan areas in Japan between 1993 and 2003, 39 (65%) of the 60 patients were male [12]. Interestingly, in the present study 19 (90.5%) of the 21 patients were male, indicating that the patients were obviously male-dominant, and seven (36.8%) of these 19 male patients were MSM. Between January and May 2018, 17 male patients infected with HAV were reported in Yokohama, Japan [23]. Of these 17 patients, 14 (82.4%) identified as MSM.

Outbreaks, and patients with severe forms of acute hepatitis A in the MSM category, have been recently reported in Japan [23,24,25] and other countries [9,10,11]. Outbreaks of acute hepatitis A, which are described in the present article, are often associated with MSM. HAV vaccination could prevent MSM from experiencing acute HAV infection, and HAV vaccination should be useful in preventing other people from being infected with HAV. It is possible that the administration of antivirals against HAV would be useful for the prevention of severe HAV infection [26,27,28,29].

We found similar isolates to the HAV subgroup 9 (S9) [17] in two non-MSM patients (no. 1 and no. 17 in Table 2). Both were non-ALF patients. In 2016 and 2017, respectively, HAV subgroup 9 (S9) was a major HAV subgroup in the Tokyo Prefecture (42.6% (20/47) and 69.0% (31/45)0, although HAV subgroup 13 (S13) was reported in only 10.6% (5/47) and 20.0% (9/45) [16]. Asakura et al. also reported that, in 2018, HAV subgroups 9 (S9) and 13 (S13) were observed in 2.6% (9/344) and 97.3% (334/344) of cases, respectively, in the Tokyo Prefecture [17]. In the present study, it may be possible that HAV subgroup 9 (S9) was not associated with MSM. Of interest, the HAV sequence obtained in one female (no. 13 in Table 2), also intermixes with those isolated from MSM group. These facts indicate that HAV subgroup 13 (S13) may be one of the main HAV variants in this area.

## 5. Conclusions

We described a recent male-dominant hepatitis A outbreak in Japan, which may be associated with MSM to some extent. These facts seem to represent recent trends in HAV infection in the Tokyo area of Japan. Antivirals against HAV, as well as HAV vaccines, may be useful for the prevention of severe HAV infection. Future studies with increased sample sizes could highlight the findings presented in the present study.

## Figures and Tables

**Figure 1 viruses-13-00207-f001:**
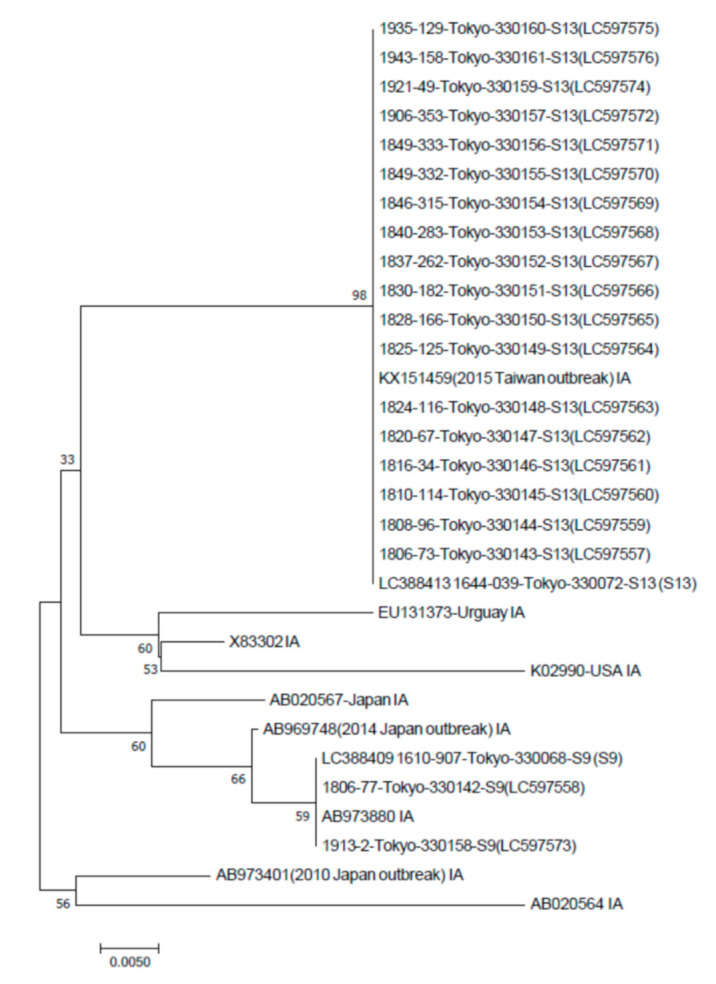
Phylogenetic analysis of the hepatitis A virus (HAV) VP1-2A region (168 bp) in the present study, and previously reported sequences of HAV subgenotype IA. Neighbor-joining methods were performed [15,17]. Accession numbers for the DDBJ/EMBL/GenBank databases are indicated. The HAV isolates in the present study ranged from LC597557 to LC597576 (see Table 2).

**Table 1 viruses-13-00207-t001:** Clinical characteristics at baseline of patients in the present study.

Characteristic	Total (*n* = 21)	ALF (*n* = 5)	Non-ALF (*n* = 16)	^1^*P*-Values
Male/female (*n*)	19/2	4/1	15/1	0.9668
Age (years)	40.7 ± 12.1	46.2 ± 5.5	39.4 ± 13.2	0.2824
BMI (kg/m^2^)	21.7 ± 3.0	21.0 ± 2.0	21.9 ± 3.3	0.5773
AST (IU/L)	3506 ± 3406	7215 ± 4228	2346 ± 2164	0.002510
ALT (IU/L)	3692 ± 2372	5789 ± 2809	3036 ± 1864	0.01913
γ-GTP (IU/L)	416 ± 199	415 ± 119	416 ± 222	0.9924
LDH (IU/L)	2002 ± 2660	4838 ± 3841	1116 ± 1410	0.00329
Albumin (g/dL)	3.9 ± 0.4	3.5 ± 0.6	3.9 ± 0.3	0.0558
Total bilirubin (mg/dL)	7.7 ± 5.1	6.7 ± 2.2	8.0 ± 5.8	0.6344
CRP (mg/dL)	1.6 ± 1.1	2.2 ± 1.6	1.4 ± 0.9	0.1665
Platelet counts (×10^4^/μL)	17.9 ± 5.4	16.8 ± 6.7	18.2 ± 5.2	0.6280
Atypical lymphocytes (/μL)	410 ± 555	644 ± 971	337 ± 367	0.2913
IgM (mg/dL)	280 ± 115	236 ± 124	291 ± 114	0.3679
MSM group (yes/no)	7/14	1/4	6/10	0.8562
Duration from onset to admission (days)	7.2 ± 4.7	4.6 ± 2.1	8.1 ± 5.0	0.1493
Duration from admission to discharge (days)	19.1 ± 9.0	20 ± 2.4	18.9 ± 10.3	0.8183

^1^*P*-Values (ALF vs. non-ALF); BMI, body mass index; AST, aspartate aminotransferase; ALT, alanine aminotransferase; γ-GTP, γ-glutamyl transpeptidase; LDH, lactate dehydrogenase; CRP, C-reactive protein; IgM, immunoglobulin M; ALF, acute liver failure; MSM, men who have sex with men.

**Table 2 viruses-13-00207-t002:** Clinical features of the 21 patients with HAV subgenotype IA infection in the present study.

Case No.	Admission Date (Year/Month)	Age (Years)/Sex	Fever/ALF/MSM	IgM-HAV (S/CO)	Duration of Admission (Days)	History of Syphilis/HBV/Eating of Raw Oysters or Fish	Subgroup/Accession Number of HAV Isolate
1	2018/January	27/male	+/−/−	1.7	22	−/−/−	S9/LC597558
2	2018/February	48/male	+/+/−	6.12	22	−/−/−	S13/LC597557
3	2018/February	56/male	−/−/−	7.47	15	−/−/+	S13/LC597559
4	2018/March	56/male	+/−/−	5.41	20	−/−/−	S13/LC597560
5	2018/April	24/male	+/−/+	8.98	20	+/−/−	S13/LC597561
6	2018/May	37/male	+/−/+	6.9	21	−/−/−	S13/LC597562
7	2018/June	39/male	−/+/+	4.61	21	+/+/−	S13/LC597563
8	2018/June	54/male	+/+/−	8.5	21	−/−/−	S13/LC597564
9	2018/July	23/male	−/−/−	7.42	37	−/−/−	S13/LC597566
10	2018/July	24/male	−/−/+	8.22	9	−/+/−	S13/LC597565
11	2018/September	44/male	+/+/−	6.26	16	−/−/+	S13/LC597567
12	2018/October	58/male	+/−/+	11.3	11	+/−/−	S13/LC597568
13	2018/November	46/female	+/+/−	3.43	20	−/−/−	S13/LC597569
14	2018/December	53/male	+/−/−	12.7	20	−/−/−	S13/LC597570
15	2018/December	32/male	−/−/−	5.25	10	−/−/−	S13/LC597571
16	2019/February	35/male	+/−/+	5.07	10	−/−/−	S13LC597572
17	2019/March	52/male	+/−/−	8.77	14	−/−/−	S9/LC597573
18	2019/May	24/male	+/−/−	7.57	14	−/−/−	S13/LC597574
19	2019/August	34/male	−/−/+	10.8	15	+/+/−	S13/LC597575
20	2019/October	38/male	−/−/−	8.98	48	−/−/+	S13/LC597576
21	2020/April	51/female	−/−/−	11.2	16	−/−/−	N/A

S/CO, sample value/cutoff value measured by chemiluminescent immunoassay (CLIA); +, positive; −, negative; HAV, hepatitis A virus; ALF, acute liver failure; MSM, men who have sex with men; HBV, hepatitis B virus; S, subgroup; N/A, not available.

**Table 3 viruses-13-00207-t003:** Clinical characteristics of men who have sex with men (MSM) and non-MSM groups of patients at the baseline in the present study.

Characteristic	MSM Group (*n* = 7)	Non-MSM Group (*n* = 14)	^1^*P*-Values
Male/female (*n*)	7/0	12/2	0.7926
Age (years)	35.9 ± 11.5	43.1 ± 12.1	0.2001
BMI (kg/m^2^)	21.4 ± 1.8	21.8 ± 3.5	0.9314
AST (IU/L)	3599 ± 3767	3458 ± 3360	0.8718
ALT (IU/L)	3569 ± 2716	3753 ± 2289	0.8718
γ-GTP (IU/L)	405 ± 212	421 ± 201	0.8675
LDH (IU/L)	2319 ± 2791	1844 ± 2686	0.7101
Albumin (g/dL)	4.0 ± 0.3	3.8 ± 0.5	0.3455
Total bilirubin (mg/dL)	5.6 ± 2.1	8.7 ± 5.9	0.1980
CRP (mg/dL)	1.3 ± 1.0	1.8 ± 1.2	0.3555
Platelet counts (×10^4^/μL)	15.5 ± 3.4	19.1 ± 6.0	0.1599
Atypical lymphocytes (/μL)	399 ± 328	416 ± 651	0.9492
IgM (mg/dL)	281 ± 91	280 ± 130	0.9857
ALF (yes/no)	1/6	4/10	0.8562
Duration from onset to admission (days)	5.4 ± 3.0	7.4 ± 5.2	0.3614
Duration from admission to discharge (days)	15.3 ± 5.4	21 ± 10.0	0.1783

^1^*P*-Values (ALF vs. non-ALF); BMI, body mass index; AST, aspartate aminotransferase; ALT, alanine aminotransferase; γ-GTP, γ-glutamyl transpeptidase; LDH, lactate dehydrogenase; CRP, C-reactive protein; IgM, immunoglobulin M; ALF, acute liver failure.

**Table 4 viruses-13-00207-t004:** Clinical characteristics between patients with admission for less than 20 days and those with admission for 20 days or more.

Characteristic	Fewer than 20 Days (*n* = 10)	20 Days or More (*n* = 11)	^1^*P*-Values
Male/female (*n*)	9/1	10/1	0.5007
Age (years)	41.0 ± 12.9	40.5 ± 12.0	0.9213
BMI (kg/m^2^)	22.2 ± 3.5	21.2 ± 2.7	0.4700
AST (IU/L)	2429 ± 3525	4484 ± 3131	0.1732
ALT (IU/L)	3090 ± 2491	4238 ± 2231	0.2790
γ-GTP (IU/L)	347 ± 118	478 ± 240	0.1351
LDH (IU/L)	1334 ± 2649	2609 ± 2643	0.2838
Albumin (g/dL)	3.8 ± 0.5	3.9 ± 0.5	0.6523
Total bilirubin (mg/dL)	7.7 ± 3.6	7.6 ± 6.4	0.9657
CRP (mg/dL)	1.4 ± 1.3	1.8 ± 1.0	0.4365
Platelet counts (×10^4^/μL)	18.0 ± 4.9	17.8 ± 6.1	0.9352
Atypical lymphocytes (/μL)	486 ± 371	342 ± 693	0.5657
IgM (mg/dL)	279 ± 99.5	281 ± 131	0.9692
ALF (yes/no)	2/8	3/8	0.9692
MSM group (yes/no)	4/6	3/8	0.5366
Duration from onset to admission (days)	7.7 ± 5.7	5.8 ± 7.0	0.5061
Duration from admission to discharge (days)	13.0 ± 2.7	24.7 ± 9.2	0.001042

^1^*P*-Values (less than 20 days vs. 20 days or more); BMI, body mass index; AST, aspartate aminotransferase; ALT, alanine aminotransferase; γ-GTP, γ-glutamyl transpeptidase; LDH, lactate dehydrogenase; CRP, C-reactive protein; IgM, immunoglobulin M; ALF, acute liver failure; MSM, men who have sex with men.

## Data Availability

All sequences generated by unbiased sequencing were submitted to DDBJ/EMBL/GenBank databases under accession numbers LC597557 to LC597576.

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
