# Peer review of "Male-Dominant Hepatitis A Outbreak Observed among Non-HIV-Infected Persons in the Northern Part of Tokyo, Japan"

_viruses, 2021, doi:10.3390/v13020207_

Round 1

Reviewer 1 Report

This study by Honda M., et al. analysed 21 HAV-infected patients collected in the northern part of Tokyo. The authors claimed that in this region of Japan, the outbreaks of HAV tend to be “male-dominant”.   This work confirms previous epidemiological studies about HAV performed by others over the years around the world that MSM have a higher risk of HAV infection. Further, the study classified, by phylogenetic analysis, HAV gt-IA sequences in two subgroups (S9 and S13), comparing the HCV-infected patients sequences with prototype and reference sequences in the analysis. Finally, the phylogenetic analysis made evident that the HAV gt-IA strain that in 2015 caused an outbreak among MSM in Taiwan was related to 18 HAV-infected patients in the cohort. Overall, this study highlights the importance of establishing prevention and response strategies (eg vaccination) specially for communities at higher risk. 

Comments: 

Line 21: double full stop

Line 23 and 24 and table 1: it could be confusing the way the non-MSM patients were considered in the study (n=14 male=12 female=2). By definition women are non-MSM, but the confusion is that the non-MSM group in the study also considered women, and not just men that are not MSM (n=7). 

Line 47: I’d recommend avoid the term “homosexual”.

Line 56: HIV-negative, instead of anti-HIV-negative and small letter in “we”.

Line 124: recently

Line 126 -127 : “the results suggest HAV infection may be a STD” Please elaborate

Author Response

Response to Reviewer 1

Thank you very much for your invaluable comments.

Response to your comments: “Line 21: double full stop”

Thank you very much for your invaluable comments.

I agree with you. Accordingly, we revised our manuscript as follows.

In Abstract section, line 24 of the revised manuscript,

…among non-HIV-infected persons in the northern part of Tokyo, Japan. Clinical information was…

Response to your comments: “Line 23 and 24 and table 1: it could be confusing the way the non-MSM patients were considered in the study (n=14 male=12 female=2). By definition women are non-MSM, but the confusion is that the non-MSM group in the study also considered women, and not just men that are not MSM (n=7).”

Thank you very much for your invaluable comments.

I agree with you. Accordingly, we revised our manuscript as follows.

In Abstract section, lines 25-27 of the revised manuscript,

…A total of 90.4% and 33.3% of patients were male and were men who have sex with men (MSM), respectively. The total bilirubin levels and platelet counts tended to be lower in the MSM group than in the non-MSM group. C-reactive…

Response to your comments: “Line 47: I’d recommend avoid the term “homosexual”.”

Thank you very much for your invaluable comments.

We agree with you. Accordingly, we revised our manuscript as follows.

In Introduction section, lines 49-50 of the revised manuscript,

…[9-11]. After 1998, sexual activity was one of the most important transmission routes of HAV infection in the metropolitan Tokyo area of Japan [12].…

Response to your comments: “Line 56: HIV-negative, instead of anti-HIV-negative and small letter in “we”.”

Thank you very much for your invaluable comments.

We agree with you. Accordingly, we revised our manuscript as follows.

In Introduction section, lines 56-57 of the revised manuscript,

…between January 2018 and April 2020. Of note, all these patients were HIV-negative; however, we recognized hepatitis A as a male-dominant disease, and identification as MSM as one of the high-risk factors for HAV infection..…

Response to your comments: “Line 124: recently”

Thank you very much for your invaluable comments.

We agree with you. Accordingly, we deleted “recently”.

Response to your comments: “Line 126 -127 : “the results suggest HAV infection may be a STD” Please elaborate.”

Thank you very much for your invaluable comments.

We agree with you. Accordingly, we deleted this part.

Reviewer 2 Report

Congratulations to the authors for coming up with a good hypothesis. Yes, the data presented here support the idea of HAV infection dominance in male and to a certain extent in the MSM subgroup. Future studies, with increased sample size, could highlight findings presented in this manuscript.

Author Response

Response to Reviewer 2

Thank you very much for your encouraging comments.

Response to your comments: “Congratulations to the authors for coming up with a good hypothesis. Yes, the data presented here support the idea of HAV infection dominance in male and to a certain extent in the MSM subgroup. Future studies, with increased sample size, could highlight findings presented in this manuscript.”

Thank you very much for your encouraging comments.

Accordingly, we revised our manuscript as follows.

In Conclusion section, lines 205-206 of the revised manuscript,

…HAV infection. Future studies with increased sample sizes could highlight the findings presented in the present study.

Round 2

Reviewer 1 Report

I have no further comments 

Author Response

Thank you for your invaluable comments.

We asked English editor to edit our manuscript again and we also submitted the certificate to Editor, Viruses.